# A COMPREHENSIVE BENCHMARK OF BATCH INTEGRATION METHODS FOR SPATIAL TRANSCRIPTOMICS USING A LARGE-SCALE VISIUM CANCER ATLAS

**Liam Ludington**[*][†] **& Xavier Secheresse**[†]
ENS Paris-Saclay
4 Avenue des Sciences, 91190 Gif-sur-Yvette, France
{liam.ludington,xavier.secheresse}@ens-paris-saclay.fr

**Khalil Ouardini**[*]**, Régis Loeb, Arthur Pignet & Vincent Cabeli**
Owkin
14 Boulevard Poissonnière, 75009 Paris, France
{khalil.ouardini,regis.loeb,arthur.pignet,vincent.cabeli}@owkin.com

**Omar Darwiche Domingues**[†]
Cohere
4 rue de Marivaux, 75002 Paris, France
omar@cohere.com

## ABSTRACT

Spatial transcriptomics (ST) enables spatially-resolved gene expression measurement, providing insights into tissue architecture and disease biology. However, batch effects from sequencing protocols, sample processing, and other technical factors can confound biological signals. Although batch correction has been extensively studied in single-cell transcriptomics, spatial integration methods lack rigorous benchmarking on large real-world datasets. This study benchmarks 11 representation-learning methods across three categories—linear, graph-based and probabilistic methods using Owkin's MOSAIC Window dataset, a large-scale spatial transcriptomics atlas of human cancers. Methods are evaluated across three criteria: batch correction, biological conservation, and spatial conservation. We also propose a new integration metric to assess robustness of representations to domain shifts and generalizability to unseen samples. Probabilistic methods (scVIVA, scVI) outperform linear and graph-based approaches in batch correction and biological conservation. On the other hand, graph-based methods excel at spatial conservation but underperform in batch integration. Out-of-Distribution (OOD) evaluation reveals that sophisticated methods show reduced performance on unseen samples while linear methods maintain robust generalization, highlighting trade-offs between integration quality and generalizability that should guide method selection for real-world applications.

## 1 INTRODUCTION

Spatial transcriptomics (ST) technologies capture mRNA counts with spatial context, complementing single-cell and bulk transcriptomics. Recent machine learning advances in ST analysis—including spatially variable gene identification, cell-cell communication, and cell type deconvolution—hold promise for biological discoveries such as new cancer subtypes, treatment response biomarkers (Jin et al., 2024; Ren et al., 2023), and tumor-immune spatial structures (Liu et al., 2023b). However, batch effects from donor, sample processing, center, and platform differ-

---

[*]Equal contribution.
[†]Work performed while at Owkin.

ences compromise the ability of machine learning methods to generate meaningful and generalizable embeddings.

The single-cell field has developed numerous batch correction methods. Linear approaches include mutual nearest neighbors and related techniques (Haghverdi et al., 2018; Shang & Zhou, 2022; Korsunsky et al., 2019), while probabilistic methods employ variational inference to disentangle biological signal from technical noise (Lopez et al., 2018; Andersson et al., 2020a; Hrovatin et al., 2024; De Donno et al., 2023; Lopez et al., 2022). For spatial data, methods combine these techniques with spatial modeling: graph-based approaches such as STAGATE (Dong & Zhang, 2022) and STAligner (Zhou et al., 2023) use graph neural networks; variational autoencoders like SIMVI (Dong et al., 2025) and scVIVA (Levy et al., 2025) apply variational inference; probabilistic models such as PRECAST (Liu et al., 2023a) optimize parametric models; and foundation models like scGPT-spatial (Wang et al., 2025) employ transformers.

Despite this proliferation, no comprehensive benchmark evaluates batch integration performance in spatial data. Existing ST benchmarks (Hu et al., 2024; Zhang & Hou, 2025) have critical limitations: they do not jointly quantify batch removal and conservation of spatial structure, lack comparison with single-cell baselines, do not evaluate how well methods generalize to unseen data (which is otherwise standard practice in machine learning), and rely on unrealistically clean datasets like DLPFC (Maynard et al., 2021) from single patients with clear boundaries—far from tumor samples with strong sample effects.

We present a rigorous benchmark evaluating 11 methods spanning linear models (PCA, fastMNN, Harmony, Combat), graph-based approaches (SEDR, STAGATE, SPIRAL), and probabilistic methods (PRECAST, scVI, scVIVA, sysVI). We measure integration quality using scIB (Luecken et al., 2022) for batch correction and biological conservation, spatial conservation metrics (Hu et al., 2024), and scGraph (Wang et al., 2024a) for geometric preservation. Our framework combines comprehensive metric suites with two validation strategies: (1) *all-at-once* integration assessing in-distribution quality, and, in contrast to previous benchmarks, (2) Out-of-Distribution (OOD) evaluation measuring generalization to held-out samples. For models that permit evaluation on unseen data, we assess generalization using an Out-of-Distribution (OOD) gap metric comparing 5-fold cross-validation performance (split stratified by donor) on held-out samples versus in-distribution integration. We open source our benchmarking library for community use.

We benchmark on Owkin's MOSAIC Window dataset (Hoffmann & Consortium, 2025b), containing samples from the MOSAIC spatial omics collection (Hoffmann & Consortium, 2025a) from multiple donors, centers, and cancer types. MOSAIC Window exhibits strong sample-specific effects (see Figure 6), providing a demanding and clinically relevant testbed.

Our results show variational inference methods outperform linear and graph-based approaches for batch integration and biological conservation. The spatially-aware scVIVA achieves best overall performance, though non-spatial methods scVI & sysVI perform comparably on many metrics. Graph-based methods excel at spatial conservation but fail at batch correction even when counts are preprocessed with Harmony, raising questions about their ability to capture biological signals under strong batch effects. These findings indicate that there remains substantial room to develop effective spatial-aware integration models. Finally our Out-of-Distribution (OOD) generalization experiments reveal underexplored tradeoffs in integration studies, as strong performers in all-at-once setting such as scVI show high sensitivity to distribution shifts, even after fine-tuning on unseen samples while linear methods such as PCA demonstrate robustness, emphasizing that method selection must consider deployment context.

All code is available at `https://github.com/owkin/st-benchmark`. Data is accessible through the MOSAIC Window viewer at `https://www.mosaic-research.com/mosaic-window`.

## 2 METHODS

### 2.1 MOSAIC WINDOW DATASET

Commonly used ST benchmarking datasets fail to reflect the complexity of clinical data. The DLPFC Visium dataset (Maynard et al., 2021) contains single-patient healthy tissue with well-

defined cortical boundaries. The HER2BT dataset (Andersson et al., 2020b) captures only 3,000 genes per spot using pre-transcrimptome-wide technology, while the Human Breast Cancer (HBC) dataset (Xu et al., 2024) contains a single slide from one HER2-positive tumor. These limitations prevent rigorous evaluation.

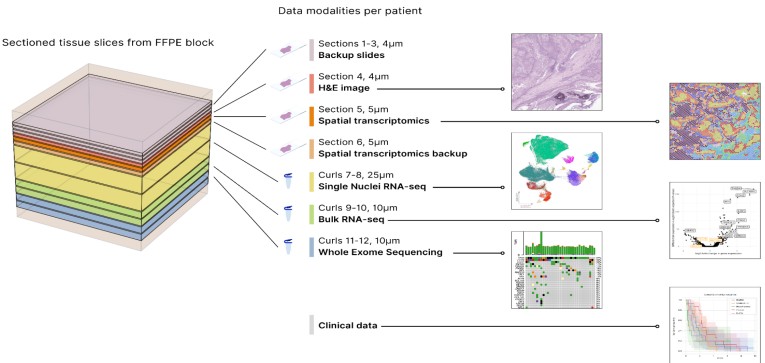

Figure 1: **Mosaic Window dataset.** Serially sectioned Formalin-Fixed Paraffin-Embedded (FFPE) tumor blocks provide: 4-5 µm sections for H&E histology and Visium spatial transcriptomics; 10-25 µm curls for single-nuclei RNA-seq, bulk RNA-seq, and whole exome sequencing. Integration with clinical data enables survival and translational analyses. Outputs show H&E morphology, spatial gene expression, single-cell clusters, genomic alterations, and survival curves.

To address this gap, we evaluate all methods on multi-sample integration tasks using the MOSAIC Window dataset. The data generation process is summarized in Figure 1. Following emergent frameworks for categorizing spatial transcriptomics batch effects (Zhang & Hou, 2025), we focus exclusively on **inter-sample batch effects**—variability arising across tissue samples from different donors. These batch effects are more obvious when integrating tissue slices from multiple samples, particularly when those samples are processed across different experimental batches. Variations in reagents, equipment, operators, or processing time can introduce systematic biases that complicate data integration. We constructed four specific cohorts, each containing about a dozen samples from different donors with the same cancer type collected at the same center — Centre Hospitalier Universitaire de Lausanne (CHUV) or Charité-Universitätsmedizin Berlin (Charité) . Each cohort's composition is summarized in Table 2.1. In these cohorts, observed batch effects (see Figure 6) emerge from technical sources (sample processing conditions, sequencing depth). Successfully integrating these cohorts requires methods to learn shared representations of cancer biology while removing sample-specific confounders.

| Center | Indication | # Samples |
|---|---|---|
| CHUV | Glioblastoma | 10 |
| CHUV | Bladder | 15 |
| Charité | Ovarian | 15 |
| CHUV | Mesothelioma | 10 |

Table 1: **Overview of the MOSAIC Window cohorts used to benchmark data integration methods**. Three of the four cohorts have the same center. Each sample originates from a different donor.

## 2.2 INTEGRATION METHODS

We evaluate 11 methods representative of three major algorithmic categories in spatial and single-cell transcriptomics: linear methods, graph-based approaches, and probabilistic models including variational inference architectures. We omit a spate of recent genomic foundation models (Wang et al., 2025) on the grounds of their relatively high computational cost and unclear advantage over existing SOTA probabilistic methods. We restrict evaluation to methods requiring only spatial mRNA counts and spot coordinates. While several existing methods leverage paired modalities—such as

GraphST (Long et al., 2023), which uses H&E images, or DestVI (Lopez et al., 2022), which uses paired single-cell data—we exclude these to ensure fair comparison.

All methods use raw counts from Space Ranger (10x Genomics, 2024) with minimal Unique Molecular Identifier (UMI) filtering as input. Method-specific preprocessing is summarized in Table 2. For all methods, we use the default hyperparameters from the original implementation. Non-integrative methods STAGATE and SEDR are preprocessed with Harmony as shown in their tutorials.

| Type | Method | Spatial | Integrative | Preprocessing | 🐍/ⓡ | Unseen |
|---|---|---|---|---|---|---|
| Linear | PCA | ✗ | ✓ | CPM + log1P + HVGs | 🐍 | ✓ |
| | fastMNN | ✗ | ✓ | CPM + log1P + HVGs | 🐍 | ✗ |
| | Harmony | ✗ | ✓ | CPM + log1P + HVGs | ⓡ | ✗ |
| | Combat | ✗ | ✓ | CPM + log1P + HVGs | ⓡ/🐍 | ✓ |
| Graph-ML | SEDR | ✓ | ✗ | CPM + log1P + HVGs + Harmony | 🐍 | ✓ |
| | STAGATE | ✓ | ✗ | CPM + log1P + HVGs + Harmony | 🐍 | ✓ |
| | SPIRAL | ✓ | ✓ | CPM + minmax + HVGs | 🐍 | ✓ |
| Prob. | scVI | ✗ | ✓ | HVGs | 🐍 | ✓ |
| | sysVI | ✗ | ✓ | CPM + log1P + HVGs | 🐍 | ✗ |
| | scVIVA | ✓ | ✓ | HVGs | 🐍 | ✗ |
| | PRECAST | ✓ | ✓ | HVGs | ⓡ | ✗ |

Table 2: **Overview of the 11 integration methods evaluated in this benchmark**. Methods are categorized by type (Linear, Graph ML, Probabilistic), spatial awareness, integrative capabilities, preprocessing requirements, implementation language, and ability to perform inference on unseen samples. Preprocessing steps include: counts per million (CPM) normalization, log(1+x) transformation (log1P), highly variable gene (HVG) selection, MinMax scaling, and Harmony batch correction. Implementation languages (Python/R) are indicated by icon.

## 2.3 EVALUATION METRICS

### 2.3.1 BATCH CORRECTION & BIOLOGICAL CONSERVATION

We evaluated integration quality using ten complementary metrics from the scIB suite (Luecken et al., 2022), grouped into two categories that capture the fundamental trade-off in data integration. *Batch correction* metrics assess how well the method removes unwanted technical variation by measuring the degree to which cells from different batches intermix in the latent space. *Biological conservation* metrics assess how well the method preserves meaningful biological structure by measuring whether biologically similar cells (same type) cluster together while biologically distinct cells (different types) remain separated. We scaled each metric and computed weighted averages within categories, yielding **scib-Batch** (batch mixing quality) and **scib-Bio** (biological structure preservation) scores. Each Visium spot captures RNA from multiple cell types. We therefore assign each spot to its *dominant* cell type — defined as the cell type with the highest predicted proportion. Spot-level deconvolution was performed using Cell2Location (Kleshchevnikov et al., 2022) with a reference matrix derived from single-cell RNA-seq data of matched tissue (see Figure 1) following the methodology detailed in (Hoffmann & Consortium, 2025a).

However, scIB does not account for geometric distortion between input and embedding spaces. We complement this with the scGraph metric (Wang et al., 2024b), which measures nearest-neighbor structure preservation, adding it to the weighted average of scIB biological conservation metrics. Detailed metric definitions are provided in the Supplementary.

### 2.3.2 SPATIAL CONSERVATION

We employ spatial conservation metrics from previous ST benchmarks (Hu et al., 2024) that quantify how well latent embeddings preserve 2D spatial relationships. These are scaled, then aggregated into a weighted *Spatial Conservation* score. See Supplementary for individual metric definitions.

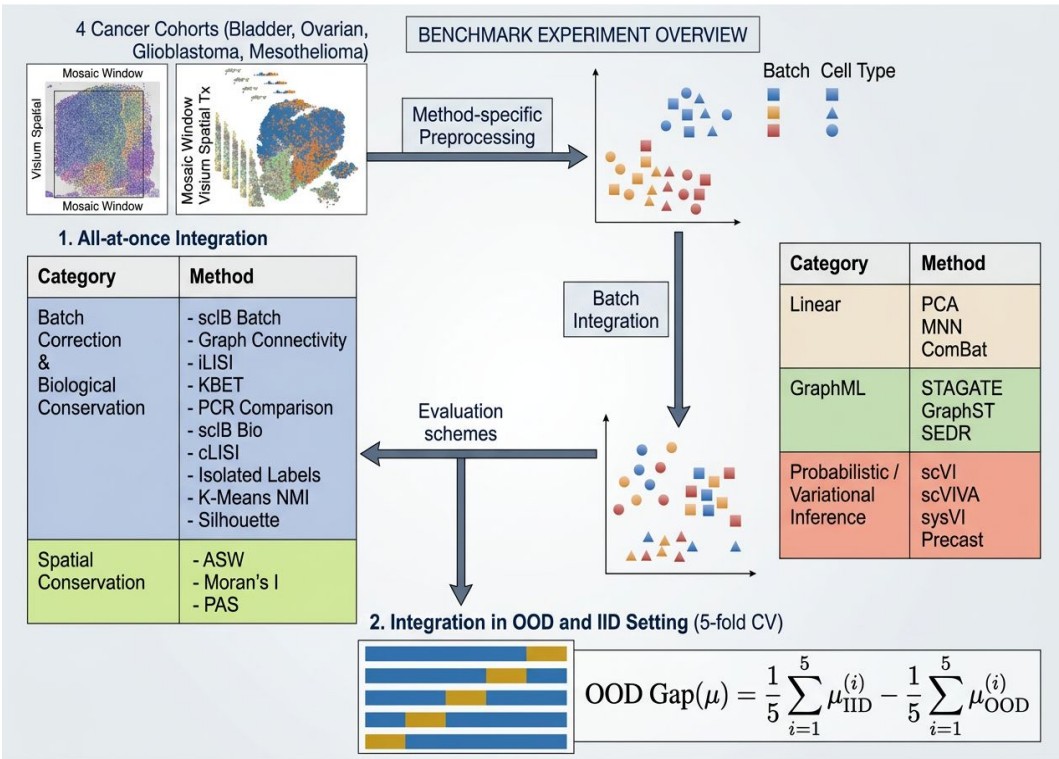

Figure 2: **Benchmark workflow**. 11 methods across three categories are evaluated on MO-SAIC Window data using *All-at-once* and In-Distribution/Out-of-Distribution (ID/OOD) validation schemes. In the *All-at-once* scheme we evaluate integration quality on the full dataset, whereas in the ID/OOD setting, we use a 5 fold cross-validation scheme and compute for each integration metric $\mu$ the difference between the average integration performance on a held out set, in OOD (unseen samples) and ID (random sampling of spots) settings, we call this metric the "OOD gap". Metrics include scIB (batch correction, biological conservation), spatial conservation, scGraph (geometric preservation)

## 3 RESULTS

### 3.1 ALL-AT-ONCE INTEGRATION

We evaluated 11 integration methods across four cohorts (Bladder, Ovarian, Glioblastoma, Mesothelioma) using three aggregated metric categories: *Batch Correction*, *Biological Conservation*, and *Spatial Conservation*. The full breakdown of the individual metrics for each cohort is available in the Supplementary. In the *all-at-once* setting, integration is evaluated on the full dataset. Figure 3 reveals consistent performance patterns independent of cancer type, suggesting that methods' strengths and weaknesses stem from algorithmic design rather than dataset-specific factors.

**Probabilistic methods lead in batch correction and biological conservation.** Variational inference methods (scVI, scVIVA, sysVI) consistently achieve superior batch correction and biological conservation across all cohorts, mirroring single-cell transcriptomics shift from linear to non-linear probabilistic approaches that explicitly model technical variation. Notably, the linear method Harmony achieves very competitive batch correction. Linear methods also achieve competitive Biological conservation performance across cohorts, attesting to their ability to preserve cell type identity in mixed transcriptomics data. While scVI and sysVI demonstrate strongest batch correction, scVIVA achieves superior biological conservation (ranks 1st on all cohorts, Figure 4) likely due to its neighborhood-aware modeling components that preserve cell type structure in local niches.

**Graph-based methods excel at spatial conservation but underperform in batch integration.** STAGATE and SPIRAL exhibit the poorest batch correction while SEDR showcases the lowest

biological conservation performance across cohorts. Poor performance persists even though counts are preprocessed with Harmony — a strong batch correction method for single-cell data, suggesting that graph-based learning that prioritizes spatial structure may degrade biological signals, limiting its utility to clustering tasks. Even SPIRAL, with domain adversarial batch integration, fails to match probabilistic methods. Local neighborhood reliance may reinforce the learning of sample-specific patterns over batch-invariant representations.

**Linear methods show task-specific strengths.** PCA demonstrates strong spatial conservation (ranks 2 on average, Figure 4), second only to STAGATE, despite being the weakest method on batch correction. This reveals that spatial patterns in cancer tissue—such as tumor-stroma boundaries and regional architecture—represent robust structural features that are determined by transcriptomic signal, suggesting spatial clustering may be inherently less challenging than batch integration in these datasets. Conversely, Harmony excels at batch correction (rank 2.25 on average, Figure 4) but shows limited spatial preservation (rank 8.50). Standard PCA's competitive spatial performance confirms that basic dimensionality reduction captures fundamental tissue organization, while its poor batch correction validates the necessity of batch-aware modeling for multi-sample integration.

**Spatial conservation reveals method specialization.** Graph methods (STAGATE, SEDR, SPIRAL) demonstrate clear spatial conservation advantages which positions them as strong candidates when downstream analyses are limited to spatial clustering tasks where batch effects are less problematic. On the other hand, methods that operate directly on count space rather than learning latent representations such as Combat show degraded spatial conservation, suggesting that the choice of representation space matters for preserving spatial relationships.

**Averaged rankings across cohorts confirm superiority of probabilistic methods.** Average rankings across cohorts (Figure 4) confirm the trends visible in Figure 3. scVIVA and scVI rank high across all categories except spatial conservation. Graph methods occupy opposite extremes: STAGATE ranks 1st spatially but last (11th) for batch correction; SEDR ranks 4th spatially but last (11th) on biological conservation. This inverse relationship suggests fundamental trade-offs in graph architectures. Linear methods cluster in the mid-to-poor range, with Harmony and PCA showing the most variability in rankings.

**scVIVA demonstrates uniquely balanced performance.** While scVI achieves superior batch correction, scVIVA uniquely ranks high on both batch correction and biological conservation while maintaining competitive spatial conservation (average rank 4.75, Figure 4). This stems from scVIVA's architecture, which extends scVI with spatial-aware components in its inference and generative networks, successfully integrating batches without sacrificing spatial relationships. This suggests that future method development should prioritize architectures that jointly model both spatial structure and batch effects within a unified probabilistic framework.

## 3.2 Out-of-Distribution Generalization

Real-world deployment sometimes requires methods to generalize to unseen samples. However, existing benchmarks rarely validate on unseen data. Many popular methods integrate all batches simultaneously (all-at-once), returning a single embedding, which is feasible for small spatial transcriptomics datasets but computationally prohibitive at scale. We used an Out-of-Distribution (OOD) evaluation framework inspired by the domain shift literature (Shi et al., 2023) to assess whether integration methods maintain performance on new samples from unseen donors.

We evaluated representative methods from each category—PCA (Linear), scVI (Probabilistic), and SEDR (Graph-based)—on the largest MOSAIC Window cohort (Bladder). SEDR used Harmony-preprocessed counts; scVI leveraged scArches (Lotfollahi et al., 2022) for inference on unseen samples. We designed a cross-validation scheme under two conditions: *In-Distribution (ID)*: evaluating integration on a held out set of spots randomly sampled from donors seen during training and *Out-of-Distribution (OOD)*: evaluating integration on a held out set of spots from unseen donors.

Our key metric, the **OOD Gap**, quantifies robustness to domain shifts by measuring the average performance difference between ID and OOD conditions (see Supplementary Methods for formal definition). A negative gap indicates performance degradation on unseen samples. The more negative the value of the gap, the more severe the degradation. Meanwhile, gaps near zero (or even positive gaps) indicate robust generalization. Despite strong in-distribution batch correction per-

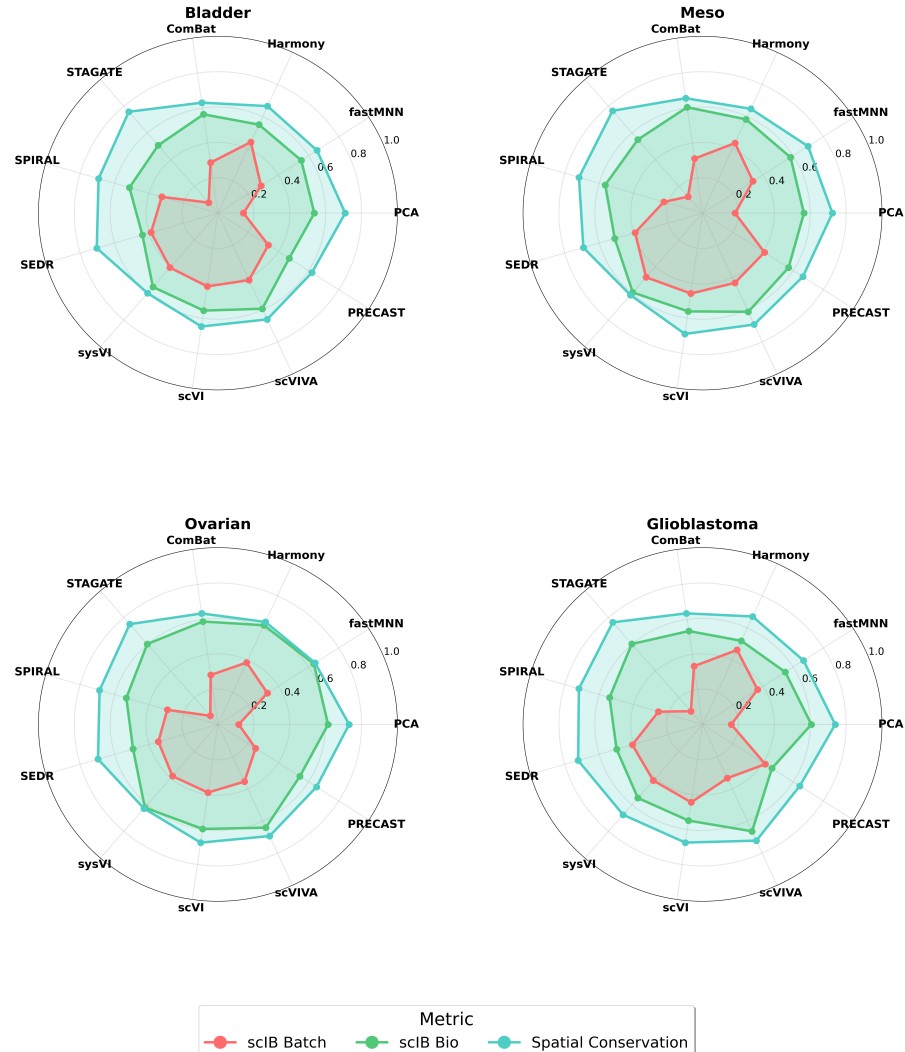

Figure 3: **Integration performance across cohorts, in *all-at-once* setting**. Radar charts show method performance across four cohorts in the *all-at-once* setting, in which integration is evaluated on the full dataset. Each vertex represents one of 11 methods; radial distance from center (0) to edge (1.0) indicates performance. Three polygons show the aggregated metrics: batch correction (red, scIB Batch), biological conservation (green, scIB Bio + scGraph), and spatial conservation (blue, ASW, Moran's I, PAS). Probabilistic models (scVI, scVIVA, sysVI) achieve best batch/biological metrics; graph methods (STAGATE, SEDR, SPIRAL) excel at spatial conservation but fail at batch integration; linear methods show intermediate performance.

formance (Figure 3), scVI exhibits substantial degradation with a mean OOD gap of $-0.27$ for batch correction—a 27 percentage point drop when generalizing to new samples. In contrast, PCA demonstrates remarkable robustness with OOD gaps near zero ($-0.03$ to $+0.05$), indicating stable performance on both seen and unseen samples. SEDR shows competitive robustness with small OOD gaps on most metrics ($-0.04$ to $+0.05$), suggesting that graph-based methods maintain stable performance on novel samples when properly initialized with Harmony processed counts. SEDR also shows the strongest spatial conservation robustness.

Substantial variability in OOD gaps across folds (error bars) highlights the importance of multi-fold evaluation. For scVI, large error bars on batch correction suggest that performance degradation varies considerably depending on which sample is held out, reflecting the high donor heterogeneity

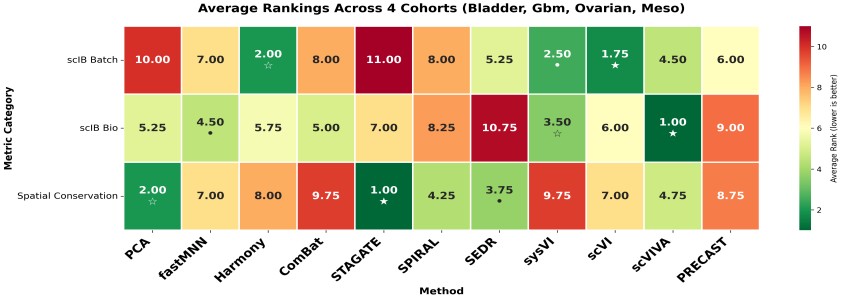

Figure 4: **Method rankings across metric categories & cohorts**. Average rankings for 11 methods across four cohorts (Bladder, Glioblastoma, Mesothelioma, Ovarian) within three categories: batch correction, biological conservation, and spatial conservation. Color indicates performance (green=best, red=worst); ranks range from 1 (best) to 11 (worst). Symbols denote top-3: ★ = rank 1, ⋆ = rank 2, • = rank 3. scVIVA and scVI dominate batch correction and biological conservation. Graph methods occupy opposite extremes: STAGATE ranks 1st spatially but last (11th) for batch correction; SEDR ranks 4th spatially but 11th biologically. This inverse relationship suggests fundamental trade-offs in graph architectures. Linear methods cluster in the mid-to-poor range, with few exceptions.

in real world settings. The full scIB Batch correction metrics breakdown shows scVI's particularly poor performance on the Principal Component Regression (PCR) metric.

These findings reveal a trade-off between integration quality and generalization. While sophisticated methods like scVI achieve superior batch correction when all samples are available during training, strong performance does not guarantee robust generalization. Simpler methods like PCA, despite lower peak performance, offer more predictable behavior on new samples. This OOD framework provides a complementary lens to traditional benchmarks, emphasizing that method selection should consider not only in-distribution performance but also deployment context and whether training data captures all the relevant technical variation that should be removed by integration methods.

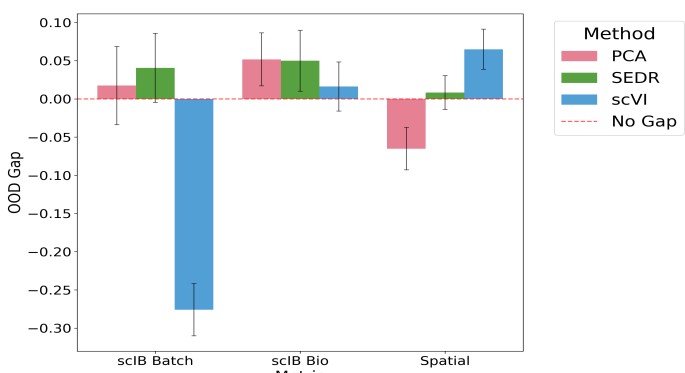

Figure 5: Averaged OOD gap across 5 folds for MOSAIC Window Bladder, for PCA, scVI and SEDR. Negative gaps indicate degradation of integration quality on unseen samples while values near zero indicate robust generalization. Error bars represent standard deviations across 5 folds.

## 4 DISCUSSION

We present one of the first comprehensive benchmark of spatial transcriptomics integration methods on large-scale, multi-donor clinical data with complex tumor biology and strong sample-specific batch effects. The MOSAIC Window dataset provides a more demanding testbed than many of the clean single-patient datasets previously used for method development.

Probabilistic methods dominate batch integration and biological conservation. Variational inference models (scVI, scVIVA, sysVI) outperform linear and graph-based methods on batch correction and biological conservation across all cohorts, mirroring single-cell transcriptomics' evolution toward probabilistic frameworks that explicitly disentangle biological signal from technical noise. However, current spatial modeling offers limited gains. Although scVIVA achieves uniquely balanced performance—ranking high in batch correction and biological conservation while maintaining competitive spatial conservation—non-spatial scVI performs comparably on many metrics with superior batch correction. This suggests that current spatial modeling approaches offer limited gains for the removal of patient-specific batch effects in clinical cohorts.

On the other hand, the benchmarked graph-based methods, STAGATE, SEDR, and SPIRAL demonstrate the strongest spatial conservation but the weakest batch correction. The low performance in biological conservation suggests that prioritizing spatial structure through graph-based learning may diminish biological signals, limiting its utility to clustering applications. Local neighborhood reliance may reinforce sample-specific patterns instead of promoting batch-invariant representations.

Finally, our generalization task on the integration of unseen samples reveals underexplored trade-offs. Our OOD evaluation reveals that sophisticated methods such as scVI achieve superior in-distribution batch correction but show substantial degradation ($-0.27$ gap) on unseen samples. Simpler methods like PCA demonstrate robust generalization (gaps near zero) despite lower peak performance, emphasizing that method selection must consider deployment context.

In conclusion, consistent performance patterns across cancer types suggest our findings generalize across tumor contexts. The MOSAIC Window dataset establishes a rigorous benchmark for evaluating progress toward unified spatial modeling and batch correction frameworks.

## MEANINGFULNESS STATEMENT

One criterion for a meaningful representation of life is that it preserves biological signal while removing technical noise, thus enabling robust generalization across diverse biological questions. As we build toward virtual cells that capture cellular behavior *in silico*, learning representations that disentangle biology from batch effects if of fundamental importance. Our benchmark reveals that probabilistic methods that model this separation—variational inference frameworks that distinguish biological variation from technical artifacts—outperform linear or graph-based approaches that conflate these signals. By establishing a rigorous evaluation framework on clinically-relevant multi-donor data, we provide insights useful for future scientists to build representations that capture biology's complexity while remaining robust to real-world deployment contexts.

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

# A SUPPLEMENTARY METHODS

## A.1 METHODS

### A.1.1 LINEAR METHODS.

Among linear methods, we have included several of the most established methods for single-cell data, as measured by scIB (Luecken et al., 2022), such as fastMNN, Harmony, and Combat:

- **PCA** Our baseline method, PCA computes the principal components of the data that account for the most variance. It then projects the data onto the principal components, giving a low-dimensional representation of the data.

- **fastMNN** Fast mutual nearest neighbors, an optimization of the original MNN algorithm, identifies mutual nearest neighbors when projecting one dataset into the other's PCA space. These spot pairs presumably stem from cells with similar biological signal and are used to determine batch effect vectors, which are then removed (Haghverdi et al., 2018).

- **Harmony** A more sophisticated take on the MNN algorithm, Harmony identifies soft clusterings of cells with a mix of K-means loss and a term that increases the batch diversity of clusters.

- **Combat** The ComBat method models observed gene counts as a linear mixture of mean gene expression, underlying biological variation, and batch effects (Johnson et al., 2007; Zhang et al., 2020). By learning the model parameters, it can regress out the batch effects from the data and return batch-corrected counts.

### A.1.2 GRAPH ML.

Among Graph ML methods, we have included SEDR for its scalability and ease-of-use, STAGATE for its performance on batch correction and clustering, and SPIRAL since it is, uniquely, inherently integrative. Since SEDR and STAGATE are not, on the other hand, inherently integrative, we preprocess their data with Harmony, as is common practice in integration studies (Zhang & Hou, 2025).

- **SEDR:** The Spatially Embedded Deep Representations (SEDR) model implements a Variational Graph Auto-encoder (VGAE) coupled with a masked reconstruction encoder-decoder framework to learn embeddings in a self-supervised manner. Embeddings are a concatenation of a gene count feature encoding module and the embedding of adjacency matrices produced by the VGAE embedding module (Xu et al., 2024).

- **STAGATE:** Another graph auto-encoder, STAGATE uses a graph attention auto-encoder framework to learn spatially-aware latents in a self-supervised manner. It employs clustering to implement cell-type aware and spatial graph attention layers (Dong & Zhang, 2022).

- **SPIRAL:** Spatial Transcriptomics Integration and Alignment (SPIRAL) uses a Graph-SAGE auto-encoder framework to learn feature representations for each spot. It additionally uses a DANN-inspired latent discriminator to integrate embeddings from different batches (Guo et al., 2023).

### A.1.3   PROBABILISTIC.

Among variational inference methods, we include scVI, which has become the *de facto* standard for single-cell data, as well as sysVI, a more recent method conceived specifically for the integration of strong batch effects, and scVIVA, a VAE recently developed for spatial transcriptomics. We have purposefully excluded methods that require additional cell-type annotations, such as scANVI or scGEN, as they have an informational advantage (cell type identity) over methods that do not require them, and because this information is not available in Visium mixed cell type data.

- **scVI:** Single-cell variational inference (scVI) utilizes a probabilistic graphical model in a variational inference framework to learn a latent representation of the data. It takes raw gene counts and batch as input, with which its inference module outputs a mean and standard deviation for the cell's latent code, assuming a normal prior. The generative network then reconstructs gene counts from the latent code and batch ID, assuming a negative binomial distribution for gene counts. The model is optimized with the standard ELBO objective (Lopez et al., 2018).

- **scVIVA:** The VAE-based scVIVA model hews closely to scVI, but makes several modeling choices to model single-cell resolution ST data. Note that this is not the case for our Visium dataset, but we adapt it to our purposes. It uses the raw gene counts, spot coordinates, and preliminary scVI embeddings to infer a latent code, which it then uses to predict reconstructed gene counts, niche composition, and niche expression.

- **sysVI:** Cross-system variational inference (sysVI) operates similarly to scVI, with two modifications meant to improve its performance on integration tasks with significant batch effects (Hrovatin et al., 2024). The first is to use a cycle consistency loss to enforce the correction of batch effects in the latent space. The second is to use a Variational Mixture of Posteriors (VAMP) prior to allow for a more expressive latent space than with a normal prior.

- **PRECAST:** Probabilistic embedding, clustering, and alignment for spatial transcriptomics (PRECAST) is a probabilistic model for integrating ST data with batch effects (Liu et al., 2023a). It assumes that observed counts are a loading matrix multiplied by the sum of a biological latent and a spatial/sample dependent batch effect. The latent codes follow a mixture of Gaussians model with mean and covariance determined by cell type, which are in turn assumed to follow a Potts model. Meanwhile, the batch effect is assumed to follow a conditional autoregressive normal distribution at each spot, allowing for batch-specific spatially cohesive batch effects.

## A.2   EVALUATION METRICS

### A.2.1   BATCH CORRECTION METRICS

Batch correction metrics assess the degree to which integration methods successfully remove technical variation while mixing cells from different batches. All metrics are scaled to the range [0, 1], where higher values indicate better batch mixing.

- **Graph Connectivity:** Measures whether cells with the same biological identity remain connected in the k-nearest neighbor (kNN) graph after integration. For each cell type, we compute the fraction of cells in the largest connected component. The final score is the average across all cell types, ranging from 0 (disconnected) to 1 (fully connected). This metric ensures batch correction does not fragment biological populations.

- **iLISI (integration Local Inverse Simpson's Index):** Quantifies batch diversity in local neighborhoods using the inverse Simpson's index. For each cell, iLISI counts how many cells can be drawn from its neighborhood before observing the same batch twice. Scores

range from 1 (no batch mixing) to B (perfect mixing, where B is the number of batches), then scaled to [0, 1]. Higher scores indicate better batch integration.

- **kBET (k-nearest-neighbor Batch Effect Test):** Tests whether the batch composition of a cell's k-nearest neighborhood matches the expected global batch composition via a $\chi^2$ test. The metric is computed separately for each cell type to account for cell-type frequency shifts across batches. The acceptance rate is averaged across cell types and scaled so that 1 indicates perfect batch mixing and 0 indicates complete batch separation.

- **PCR (Principal Component Regression):** Quantifies batch effect removal by measuring how much variance in principal components is explained by batch variables. For each PC, we compute $R^2$ from linear regression of the batch variable onto that PC, then weight by the variance explained by that PC. The total variance contribution from batch effects is summed across all PCs and subtracted from 1, such that higher scores indicate better batch removal.

- **Silhouette Batch:** Measures batch mixing using the silhouette coefficient on batch labels. For each cell, the silhouette width compares within-batch to between-batch distances. To account for biological structure, we compute batch silhouette scores separately within each cell type, then average across cell types. Scores are transformed such that 1 indicates perfect batch mixing (silhouette near 0) and 0 indicates strong batch separation.

**sciB Batch** is defined as the weighted average of the scaled individual metrics

### A.3 BIOLOGICAL CONSERVATION METRICS

Biological conservation metrics assess whether integration methods preserve meaningful biological variation. All metrics are scaled to [0, 1], where higher values indicate better conservation of biological structure.

- **cLISI (cell-type Local Inverse Simpson's Index):** Measures cell-type purity in local neighborhoods using the inverse Simpson's index. For each cell, cLISI quantifies how many cells can be drawn from its neighborhood before observing the same cell type twice. Scores range from 1 (maximum mixing) to C (perfect separation, where C is the number of cell types), then inversely scaled to [0, 1] such that higher values indicate better cell-type separation.

- **Isolated Label Scores:** Evaluates preservation of rare cell types that appear in few batches. We identify isolated labels as those present in the fewest batches, then compute two metrics: (1) F1 score of the best cluster assignment for isolated labels across multiple clustering resolutions, and (2) silhouette width comparing isolated to non-isolated labels. The final score is the mean across both metrics and all isolated labels, scaled to [0, 1].

- **K-means NMI (Normalized Mutual Information):** Compares the overlap between cell-type labels and clusters obtained from k-means clustering on integrated data. NMI measures the mutual information between the two partitions, normalized by their entropies, ranging from 0 (no overlap) to 1 (perfect match). We optimize clustering resolution (0.1 to 2.0) to find the best match with cell-type labels.

- **Silhouette Label:** Measures cell-type separation using the silhouette coefficient on cell-type labels. For each cell, the silhouette width compares within-cell-type to between-cell-type distances. The average silhouette width is computed across all cells and scaled to [0, 1], where higher values indicate well-separated, cohesive cell-type clusters.

- **scGraph:** Quantifies geometric preservation by measuring the similarity between the k-nearest neighbor structure of the input data and the integrated embeddings (Wang et al., 2024b). The metric computes the Jaccard similarity between neighborhood sets before and after integration, averaged across all cells. Scores range from 0 (no preservation) to 1 (perfect preservation), with higher values indicating that integration maintains the local geometry of the original data.

**sciB Bio** is defined as the weighted average of the scaled individual metrics, including **scGraph**.

### A.3.1 SPATIAL CONSERVATION METRICS

- **Average Silhouette Width:** finds the center of mass of each latent cluster and computes the average distance from the center of mass to every point in that latent cluster. Also computes the distance from the center of mass to the closest point in a different latent cluster. The average silhouette width is a function of these two distances. A negative ASW indicates that the cluster's CoM is closer to at least one spot in a different latent cluster than spots in its own cluster on average.
- **Moran's I:** measures spatial auto-correlation of embeddings. It is calculated like standard correlation but weighted by a spatial kernel that weights spatially close spots high and distant spots low. Moran's I ranges from -1 to 1, with 1 indicating perfect positive spatial auto-correlation.
- **PAS:** measures the percentage of spots whose majority of spatial neighbors belong to different latent-space clusters.

**Spatial conservation** is defined as the average of the scaled individual spatial metrics.

### A.3.2 METRIC SCALING

To ensure reproducibility and enable cross-study comparisons, all metrics are scaled using fixed, domain-appropriate ranges rather than observed ranges across compared methods. For spatial conservation metrics, we use theoretically motivated bounds: Moran's I and ASW use $[-1, 1]$ based on their correlation and silhouette coefficient definitions, while CHAOS and PAS use $[0, 1]$ as probability- and distance-based metrics. All scIB metrics are pre-normalized to $[0, 1]$ by design (Luecken et al., 2022). This fixed-range approach prevents data leakage, where a method's score would otherwise depend on which other methods are included in the comparison. For any metric lacking a predefined range, we fall back to observed ranges with an explicit warning to ensure transparency.

### A.4 OUT-OF-DISTRIBUTION EVALUATION FRAMEWORK

Our key metric, the **OOD Gap**, quantifies robustness to domain shifts by measuring the average performance difference between ID and OOD conditions across 5 folds for a given integration metric $\mu$:

$$\text{OOD Gap}(\mu) = \frac{1}{5}\sum_{i=1}^{5}\mu_{\text{ID}}^{(i)} - \frac{1}{5}\sum_{i=1}^{5}\mu_{\text{OOD}}^{(i)} \tag{1}$$

where $\mu_{\text{ID}}^{(i)}$ and $\mu_{\text{OOD}}^{(i)}$ represent the integration metric in ID and OOD settings for fold $i$. A negative gap indicates performance degradation on unseen samples, while a gap near zero indicates robust generalization.

## B SUPPLEMENTARY RESULTS

Table 3: Integration method performance on Bladder cohort. Best performance for each metric is highlighted in bold.

| Metric | PCA | fastMNN | Harmony | ComBat | STAGATE | SPIRAL | SEDR | sysVI | scVI | scVIVA | PRECAST |
|---|---|---|---|---|---|---|---|---|---|---|---|
| *Batch Correction* | | | | | | | | | | | |
| Graph Connectivity | 0.495 | 0.529 | 0.517 | 0.043 | 0.244 | 0.486 | 0.396 | 0.590 | 0.595 | **0.724** | 0.239 |
| iLISI | 0.000 | 0.048 | 0.076 | 0.050 | 0.000 | 0.000 | 0.022 | 0.154 | 0.079 | 0.049 | **0.226** |
| KBET | 0.074 | 0.159 | **0.289** | 0.062 | 0.070 | 0.094 | 0.147 | 0.135 | 0.217 | 0.206 | 0.142 |
| PCR Comparison | 0.000 | 0.406 | 0.885 | **1.000** | 0.000 | 0.725 | 0.988 | 0.751 | 0.783 | 0.688 | 0.730 |
| *Batch Correction (Agg.)* | 0.142 | 0.286 | **0.442** | 0.288 | 0.079 | 0.326 | 0.388 | 0.408 | 0.419 | 0.416 | 0.334 |
| *Bio Conservation* | | | | | | | | | | | |
| cLISI | 0.997 | 0.989 | 0.974 | 0.989 | 0.968 | 0.987 | 0.955 | 0.976 | 0.976 | **0.997** | 0.927 |
| Isolated Labels | 0.373 | 0.385 | 0.388 | 0.425 | 0.365 | 0.396 | 0.304 | 0.372 | 0.473 | **0.484** | 0.401 |
| K-Means NMI | 0.310 | 0.347 | 0.328 | **0.353** | 0.245 | 0.176 | 0.132 | 0.340 | 0.278 | 0.352 | 0.162 |
| Silhouette Label | 0.468 | 0.486 | 0.512 | 0.490 | 0.453 | 0.496 | 0.364 | 0.524 | 0.501 | **0.545** | 0.399 |
| scGraph | 0.830 | 0.857 | **0.898** | 0.824 | 0.802 | 0.694 | 0.521 | 0.791 | 0.644 | 0.669 | 0.714 |
| *Bio Conservation (Agg.)* | 0.537 | 0.552 | 0.550 | 0.564 | 0.508 | 0.514 | 0.439 | 0.553 | 0.557 | **0.595** | 0.472 |
| *Spatial Metrics* | | | | | | | | | | | |
| ASW | -0.110 | -0.193 | -0.080 | -0.247 | **0.084** | -0.228 | -0.200 | -0.259 | -0.231 | -0.260 | -0.182 |
| Moran's I | 0.377 | 0.367 | 0.349 | 0.112 | 0.467 | 0.403 | **0.477** | 0.399 | 0.370 | 0.416 | 0.382 |
| PAS | 0.006 | 0.119 | 0.138 | 0.039 | **0.000** | 0.009 | 0.028 | 0.274 | 0.127 | 0.096 | 0.232 |
| *Spatial Conservation (Agg.)* | 0.709 | 0.656 | 0.665 | 0.631 | **0.758** | 0.693 | 0.704 | 0.599 | 0.648 | 0.661 | 0.623 |

Table 4: Integration method performance on Glioblastoma cohort. Best performance for each metric is highlighted in bold.

| Metric | PCA | fastMNN | Harmony | ComBat | STAGATE | SPIRAL | SEDR | sysVI | scVI | scVIVA | PRECAST |
|---|---|---|---|---|---|---|---|---|---|---|---|
| *Batch Correction* | | | | | | | | | | | |
| Graph Connectivity | 0.530 | 0.707 | 0.682 | 0.064 | 0.394 | 0.540 | 0.521 | 0.789 | 0.800 | **0.899** | 0.394 |
| iLISI | 0.000 | 0.065 | 0.091 | 0.095 | 0.000 | 0.000 | 0.007 | 0.164 | 0.073 | 0.021 | **0.260** |
| KBET | 0.111 | 0.207 | 0.184 | 0.173 | 0.003 | 0.111 | 0.109 | 0.211 | 0.166 | 0.137 | **0.266** |
| PCR Comparison | 0.000 | 0.480 | 0.894 | **1.000** | 0.000 | 0.374 | 0.994 | 0.513 | 0.740 | 0.278 | 0.744 |
| *Batch Correction (Agg.)* | 0.160 | 0.365 | **0.463** | 0.333 | 0.099 | 0.256 | 0.407 | 0.419 | 0.445 | 0.334 | 0.416 |
| *Bio Conservation* | | | | | | | | | | | |
| cLISI | 0.999 | 0.982 | 0.962 | 0.990 | 0.987 | 0.993 | 0.960 | 0.955 | 0.965 | **1.000** | 0.887 |
| Isolated Labels | 0.579 | 0.466 | 0.454 | 0.460 | 0.592 | 0.514 | 0.471 | 0.427 | 0.491 | **0.632** | 0.392 |
| K-Means NMI | 0.342 | 0.269 | 0.204 | 0.208 | 0.335 | 0.164 | 0.082 | 0.320 | 0.261 | **0.455** | 0.191 |
| Silhouette Label | 0.505 | 0.475 | 0.460 | 0.477 | 0.493 | 0.489 | 0.482 | 0.499 | 0.480 | **0.571** | 0.368 |
| scGraph | 0.790 | **0.836** | 0.768 | 0.770 | 0.778 | 0.770 | 0.727 | 0.756 | 0.603 | 0.746 | 0.722 |
| *Bio Conservation (Agg.)* | 0.606 | 0.548 | 0.520 | 0.534 | 0.602 | 0.540 | 0.499 | 0.550 | 0.549 | **0.664** | 0.459 |
| *Spatial Metrics* | | | | | | | | | | | |
| ASW | 0.063 | -0.153 | -0.088 | -0.266 | **0.117** | -0.081 | -0.113 | -0.055 | -0.143 | -0.043 | -0.089 |
| Moran's I | 0.383 | 0.366 | 0.352 | 0.111 | **0.468** | 0.398 | 0.468 | 0.403 | 0.390 | 0.418 | 0.360 |
| PAS | 0.005 | 0.100 | 0.118 | 0.018 | **0.000** | 0.005 | 0.010 | 0.147 | 0.099 | 0.021 | 0.204 |
| *Spatial Conservation (Agg.)* | 0.739 | 0.669 | 0.671 | 0.635 | **0.764** | 0.718 | 0.722 | 0.676 | 0.675 | 0.722 | 0.644 |

Table 5: Integration method performance on Mesothelioma cohort. Best performance for each metric is highlighted in bold.

| Metric | PCA | fastMNN | Harmony | ComBat | STAGATE | SPIRAL | SEDR | sysVI | scVI | scVIVA | PRECAST |
|---|---|---|---|---|---|---|---|---|---|---|---|
| *Batch Correction* | | | | | | | | | | | |
| Graph Connectivity | 0.563 | 0.773 | 0.706 | 0.121 | 0.390 | 0.570 | 0.465 | 0.820 | 0.783 | **0.848** | 0.380 |
| iLISI | 0.000 | 0.002 | 0.028 | 0.008 | 0.000 | 0.000 | 0.013 | 0.119 | 0.019 | 0.005 | **0.246** |
| KBET | 0.162 | 0.237 | 0.215 | 0.117 | 0.104 | 0.023 | 0.097 | **0.278** | 0.215 | 0.149 | 0.221 |
| PCR Comparison | 0.000 | 0.324 | 0.793 | **1.000** | 0.000 | 0.306 | 0.992 | 0.702 | 0.820 | 0.732 | 0.795 |
| *Batch Correction (Agg.)* | 0.181 | 0.334 | 0.435 | 0.311 | 0.123 | 0.225 | 0.392 | **0.480** | 0.459 | 0.434 | 0.410 |
| *Bio Conservation* | | | | | | | | | | | |
| cLISI | 0.999 | 0.996 | 0.994 | **1.000** | 0.984 | 0.995 | 0.963 | 0.991 | 0.989 | 0.999 | 0.961 |
| Isolated Labels | **0.500** | 0.500 | 0.500 | 0.500 | 0.500 | 0.500 | 0.500 | 0.500 | 0.500 | 0.500 | 0.500 |
| K-Means NMI | 0.267 | 0.334 | 0.306 | **0.399** | 0.247 | 0.248 | 0.145 | 0.327 | 0.303 | 0.381 | 0.250 |
| Silhouette Label | 0.493 | 0.502 | 0.533 | 0.521 | 0.470 | 0.517 | 0.431 | 0.555 | 0.454 | **0.574** | 0.569 |
| scGraph | 0.789 | 0.836 | 0.839 | **0.839** | 0.785 | 0.728 | 0.665 | 0.733 | 0.606 | 0.643 | 0.754 |
| *Bio Conservation (Agg.)* | 0.565 | 0.583 | 0.583 | 0.605 | 0.550 | 0.565 | 0.510 | 0.593 | 0.561 | **0.613** | 0.570 |
| *Spatial Metrics* | | | | | | | | | | | |
| ASW | -0.018 | -0.085 | -0.139 | -0.136 | **0.127** | -0.097 | -0.219 | -0.204 | -0.140 | -0.198 | -0.128 |
| Moran's I | 0.365 | 0.349 | 0.312 | 0.099 | **0.467** | 0.410 | 0.455 | 0.396 | 0.379 | 0.421 | 0.376 |
| PAS | 0.003 | 0.035 | 0.141 | 0.012 | **0.000** | 0.004 | 0.042 | 0.256 | 0.049 | 0.034 | 0.130 |
| *Spatial Conservation (Agg.)* | 0.724 | 0.699 | 0.648 | 0.656 | **0.765** | 0.717 | 0.692 | 0.613 | 0.690 | 0.693 | 0.665 |

Table 6: Integration method performance on Ovarian cohort. Best performance for each metric is highlighted in bold.

| Metric | PCA | fastMNN | Harmony | ComBat | STAGATE | SPIRAL | SEDR | sysVI | scVI | scVIVA | PRECAST |
|---|---|---|---|---|---|---|---|---|---|---|---|
| *Batch Correction* | | | | | | | | | | | |
| Graph Connectivity | 0.410 | 0.611 | 0.563 | 0.029 | 0.261 | 0.464 | 0.394 | 0.659 | 0.651 | **0.792** | 0.304 |
| iLISI | 0.000 | 0.032 | 0.044 | 0.081 | 0.000 | 0.000 | 0.001 | 0.124 | 0.038 | 0.013 | **0.190** |
| KBET | 0.054 | **0.193** | 0.097 | 0.021 | 0.000 | 0.060 | 0.000 | 0.113 | 0.189 | 0.188 | 0.052 |
| PCR Comparison | 0.000 | 0.476 | 0.835 | **1.000** | 0.000 | 0.652 | 0.988 | 0.650 | 0.678 | 0.429 | 0.453 |
| *Batch Correction (Agg.)* | 0.116 | 0.328 | 0.385 | 0.283 | 0.065 | 0.294 | 0.346 | 0.387 | **0.389** | 0.355 | 0.249 |
| *Bio Conservation* | | | | | | | | | | | |
| cLISI | **1.000** | 0.999 | 0.997 | 0.984 | 0.990 | 0.996 | 0.980 | 0.994 | 0.995 | 1.000 | 0.958 |
| Isolated Labels | 0.586 | 0.566 | 0.534 | 0.448 | **0.598** | 0.458 | 0.439 | 0.564 | 0.560 | 0.536 | 0.363 |
| K-Means NMI | 0.353 | 0.420 | 0.370 | 0.396 | 0.319 | 0.166 | 0.111 | 0.378 | 0.351 | **0.453** | 0.290 |
| Silhouette Label | 0.515 | 0.554 | 0.567 | 0.524 | 0.499 | 0.508 | 0.441 | 0.552 | 0.482 | **0.577** | 0.560 |
| scGraph | 0.856 | 0.906 | 0.893 | **0.916** | 0.848 | 0.619 | 0.654 | 0.830 | 0.574 | 0.469 | 0.739 |
| *Bio Conservation (Agg.)* | 0.614 | 0.635 | 0.617 | 0.588 | 0.602 | 0.532 | 0.493 | 0.622 | 0.597 | **0.642** | 0.543 |
| *Spatial Metrics* | | | | | | | | | | | |
| ASW | -0.004 | -0.245 | -0.145 | -0.288 | **0.025** | -0.254 | -0.269 | -0.193 | -0.258 | -0.214 | -0.146 |
| Moran's I | 0.399 | 0.388 | 0.364 | 0.127 | 0.478 | 0.399 | **0.482** | 0.423 | 0.395 | 0.430 | 0.411 |
| PAS | 0.004 | 0.141 | 0.198 | 0.017 | **0.000** | 0.011 | 0.017 | 0.229 | 0.043 | 0.027 | 0.173 |
| *Spatial Conservation (Agg.)* | 0.731 | 0.644 | 0.637 | 0.634 | **0.750** | 0.687 | 0.696 | 0.629 | 0.675 | 0.694 | 0.653 |

## C  SUPPLEMENTARY FIGURES

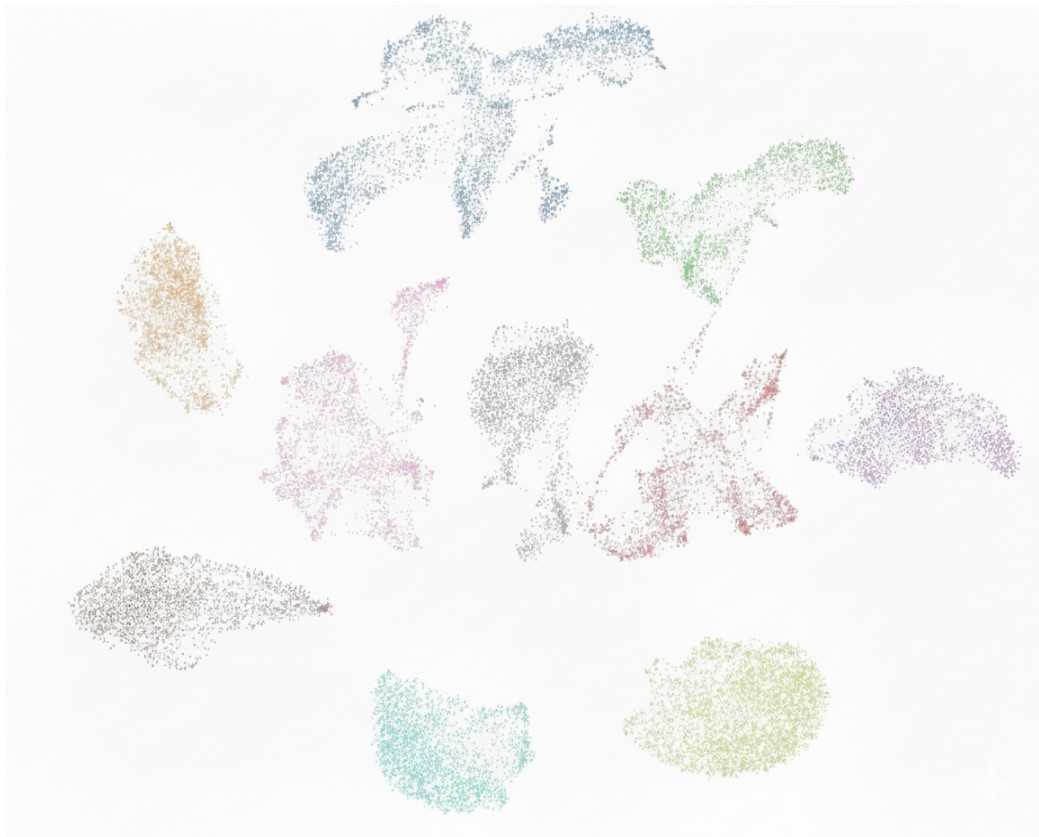

Figure 6: **Batch effects in uncorrected spatial transcriptomics counts of MOSAIC Mesothelioma samples**. UMAP visualization of Mesothelioma samples from the MOSAIC Window dataset without batch correction (each color is a different sample). Data was preprocessed using CPM normalization, log transformation, highly variable gene (HVG) selection, and PCA. Each color represents a distinct sample. Strong sample-specific clustering demonstrates substantial batch effects that confound biological signal in the absence of integration methods.

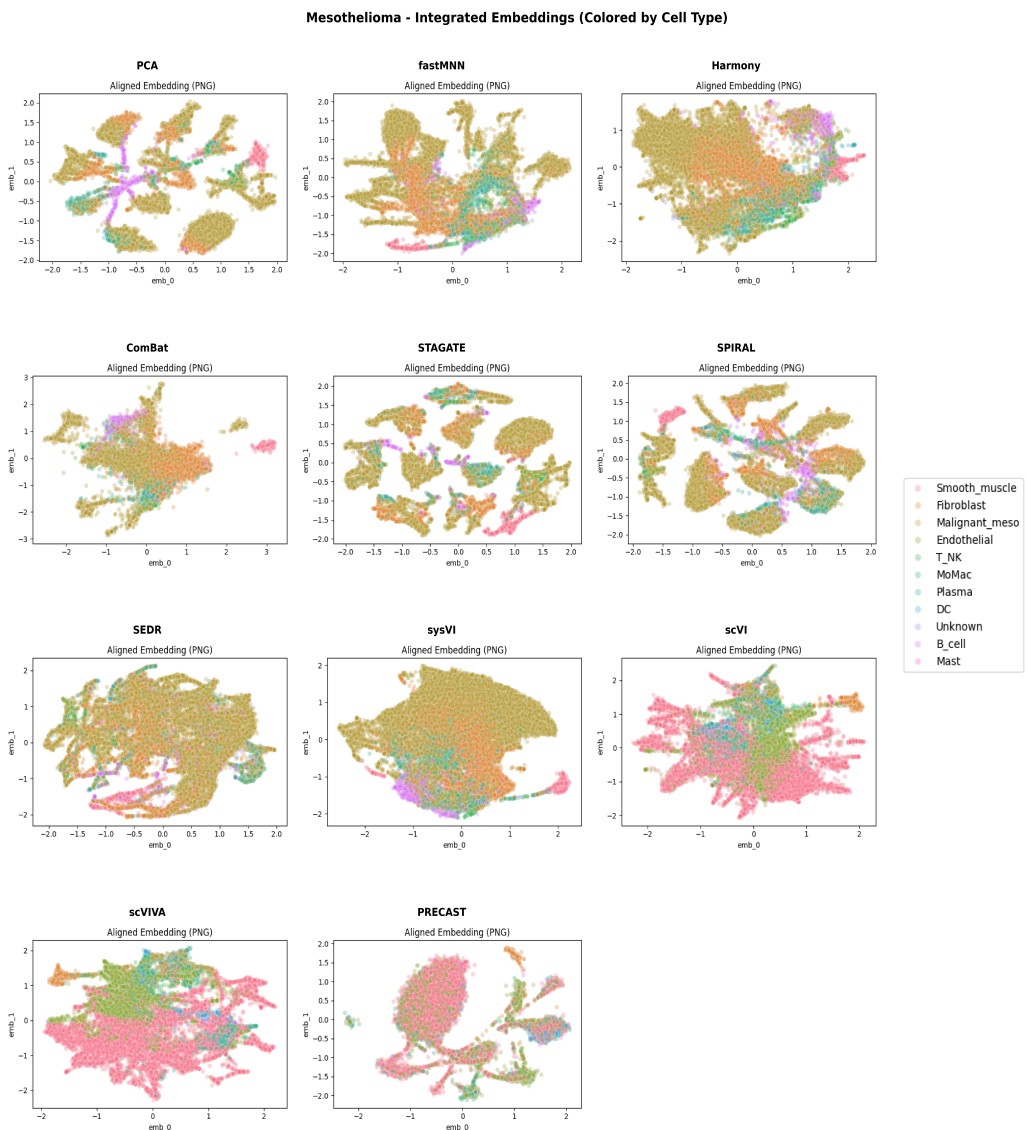

Figure 7: **Integrated embeddings colored by cell type across methods in MOSAIC Window Mesothelioma samples**. UMAP visualizations of integrated embeddings from 11 representation-learning methods applied to the MOSAIC Window Mesothelioma cohort, colored by dominant cell type annotation (i.e the cell type with the dominant proportion after spot-level deconvolution). Methods evaluated include four linear approaches (PCA, fastMNN, Harmony, Combat), three graph-based methods (STAGATE, SPIRAL, SEDR), and four probabilistic models (scVI, scVIVA, sysVI, PRECAST). Cell types shown include Malignant, Macrophages (MoMac), Smooth muscle, Endothelial, T/NK, Granulocyte, Oligodendrocyte, and Neuron.

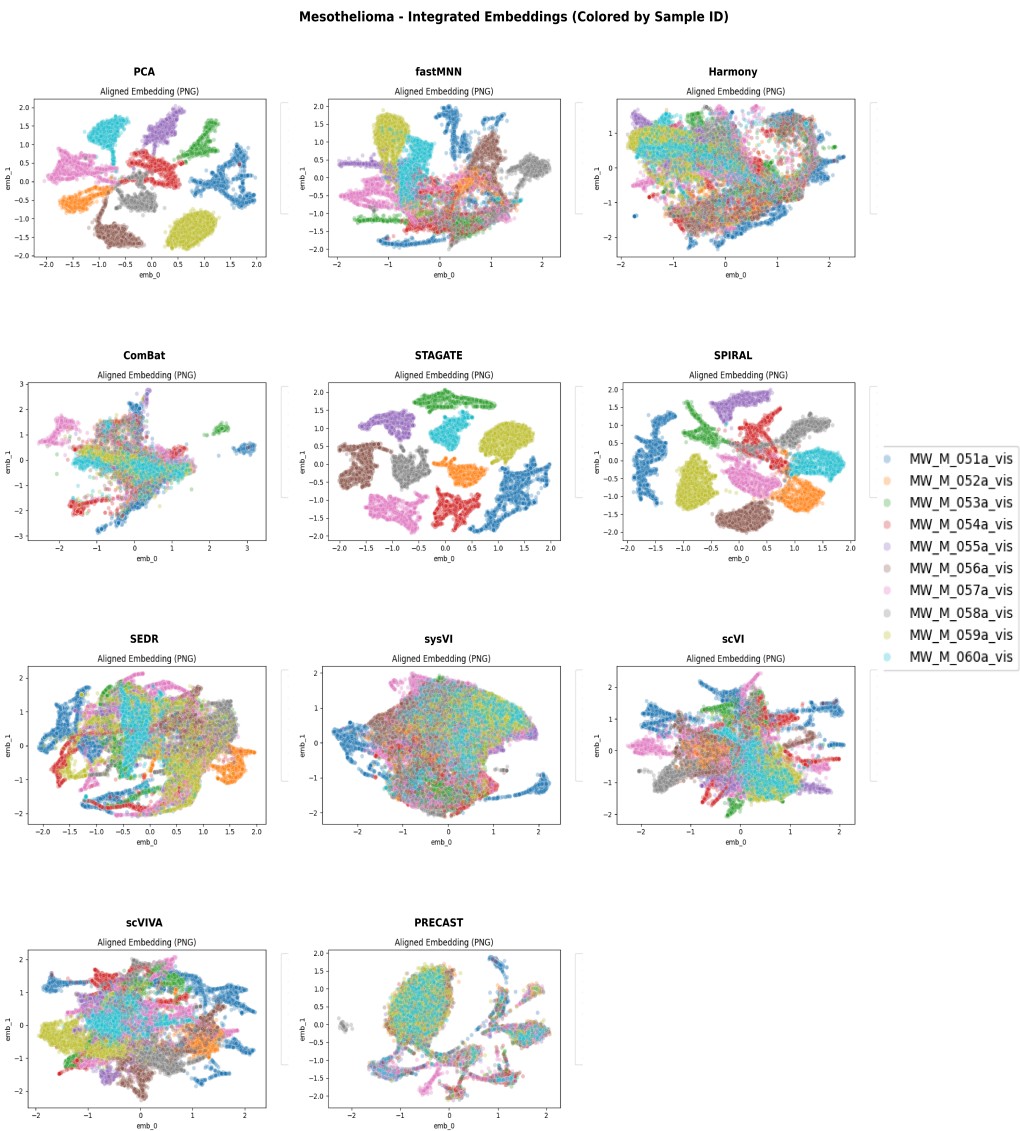

Figure 8: **Integrated embeddings colored by sample ID across methods in MOSAIC Window Mesothelioma samples.**. UMAP visualizations of integrated embeddings from the same 11 methods shown in Figure 2, colored by sample ID. Each color represents a distinct sample in the MOSAIC Window Mesothelioma cohort. Visual comparison with Figure 2 illustrates the relationship between sample mixing and biological structure preservation across different integration approaches.

