# OpenReview forum: "A Comprehensive Benchmark of Batch Integration Methods for Spatial Transcriptomics Using a Large-Scale Cancer Atlas"
_ICLR.cc/2026/Workshop/LMRL — ICLR 2026 Workshop LMRL Poster_

### Official Review · Reviewer_vJ3D · 2026-02-16
**Potentially comprehensive ST benchmark, but critical methodological gaps limit robustness and significance of the findings**

**Rating:** 2
**Confidence:** 4

**Review:**

This paper presents a benchmarking study of 11 representation-learning methods for spatial transcriptomics (ST), categorized into linear, graph-based, and probabilistic. Each method is evaluated across three criteria batch correction, biological conservation, and spatial preservation to validate both in-distribution and out-of-distribution quality. The authors define their main contribution as (1) presenting the first comprehensive benchmark of ST data on large scale multi-donor data with complex tumor and strong sample specific batch effects; and (2) introducing a new integration metric OOD Gap, which measures the robustness to generalize to unseen samples.

Strength:
- The paper identifies the need for a comprehensive benchmark for batch integration performance in spatial transcriptomics data.
- Comparison of 4 linear, 3 graph-based, and 4 probabilistic methods.
- Methods are assessed using a combination of established scIB metrics for batch and biological conservation, spatial-specific metrics, and the OOD evaluation framework

Weak points:
- All methods are compared without specifying how the hyperparameters were chosen. Were default values used? Many of these models are likely very sensitive to hyperparameters (e.g., number of principal components, learning rate, number of neighbors, etc.)
- Hyperparameters should be optimized (at least the most important ones) using the training set, and performance should be reported on the test set—ideally via cross-validation or multiple splits
- Only description of the preprocessing step, but without information on exact hyperparameters (e.g., normalisation constant, #HVG, etc)
=> This critical omission undermines the robustness and significance of the findings, particularly in a machine learning context.

- Line 65 ff.: How was the data split for 5-fold CV? In particular, as each cohort might contain samples from multiple donors, replicates etc.
- Line 70 ff.: What evidence (e.g. metrics) supports the claim that MOSAIC Window has stronger sample-specific effects than standard benchmarks?
- Table 1: Could you please clarify the steps for SEDR and STAGATE? Specifically, whether Harmony was applied as a preprocessing step to the gene expression matrix or to the SEDR and STAGATE embeddings afterwards, as shown in their tutorials.
- Line 174 ff. & Appendix A.2.1/A.3: How are the individual metrics weighted to calculate the weighted averages for scIB Batch and scIB Bio?

Minor Comments:
- I noticed some inconsistencies in how methods are named across the paper. For example, Figure 2 uses “Stagate”, but Table 1 uses “STAGATE”.
- Figure 2: "Harmony" is listed in Table 1 but seems to be missing from the batch integration table in Figure 2.

---

### Official Review · Reviewer_AG5n · 2026-02-20
**A timely benchmark of batch integration methods for spatial transcriptomics, but with limited methodological novelty, omissions of recent foundation models, and presentation issues.**

**Rating:** 4
**Confidence:** 3

**Review:**

Summary:

The authors present a benchmark of 11 batch integration methods (categorized as linear, graph-based and probabilistic) tested on spatial transcriptomics data (4 tissue types of the MOSAIC windows dataset). They identifiy scVIVA as performing overall best. Further, they test the out-of-distribution generalization ability of 3 batch integration methods.

Pros:
- batch integration for ST is a timely and relevant problem and a benchmark comparing recent methods would be useful for the community
- benchmark shows consistent trends across 4 different organs

Cons:
- The novelty of the work is limited: it uses an existing dataset (MOSAIC Window) as well as existing benchmark tasks scRNA-seq (scIB Luecken et al., 2022) and ST (Hu et al., 2024).
- The benchmark does not cover recent foundation model approaches for scRNA-seq and ST data (e.g., scGPT(-spatial) or Nicheformer), which would strengthen its relevance and actuality.
- I have concerns about the meaningfulness of the spatial conversation tasks/metrics (AWS, Moran's I, PAS). These seems to all measure the continuity of the latent embeddings with respect to their 2D spot locations. However, whether such continuity is actually favorable should depend on the true biological continuity/heterogeneity. For instance, in an immune-cold tumor one would expect less spatial continuity than in an immune-infiltrated tumor. The proposed metrics however only measure the continuity of the representations in absolute terms but not relative to the true biological heterogeneity. On this absolute scale it is not clear which level/direction of the metrics would actually be optimal. This ambiguity is further compounded by averaging these three metrics into one score.
- The description of the benchmark methods are at times vague and would benefit from indicating precise math formulas (e.g. line 725: what does '[...] compares it to the clostest [...]" mean exactly?). Moreover, hyperparameters (e.g. #clusters) are not indicated.
- The performance of the tested methods might depend on the used hyperparameters (e.g. #prinical components, #latent dimensions etc.). I cannot find an specification of these nor justifcation how they were chosen.
- The quality of Figure 2 does not meet minimal standards: The UMAP seems to be pasted in multiple times by accident, the Methods table lists only 10 of the 11 tested methods, the layout is difficult to parse, steps are enumerated in inconsistenly (4,nan,nan,1,2) and texts are underlined in red due to spell checking (and copy-pasting of the figure as a screenshot?).
- The out-of-distribution experiment was performed only on 1 out of 4 tissue types with 3 of the 11 batch integratiom methods, limiting its significance. Conclusions drawn for the entire method class (e.g. line 358 "suggesting that graph-based methods maintain stable performance on novel samples" are not justified by these.

---

### Official Review · Reviewer_Rodx · 2026-02-20

**Rating:** 7
**Confidence:** 4

**Review:**

This paper dives into a thorough benchmark of 11 different spatial transcriptomics (ST) integration methods, categorized into linear, graph-based, and probabilistic approaches. By leveraging the extensive, multi-donor MOSAIC Window cancer dataset, the authors assess key factors like batch correction, biological conservation, and spatial conservation. A standout feature of this work is the introduction of an Out-of-Distribution (OOD) generalization framework, which evaluates how well these methods hold up against domain shifts. The findings show that while probabilistic methods excel in performance with known data, simpler linear methods tend to generalize more effectively when faced with new, unseen samples.

Quality:

The empirical quality of this benchmark is outstanding.

1. Unlike previous benchmarks in spatial transcriptomics that focus on single-patient tissues with clear boundaries (like the DLPFC), this study takes a different approach by using the MOSAIC Window dataset, which includes four diverse cancer cohorts: Bladder, Ovarian, Glioblastoma, and Mesothelioma. This choice creates a much more realistic and clinically relevant testing ground, marked by significant sample-specific batch effects.

2. The evaluation process is thorough, assessing batch mixing quality and the preservation of biological structures with the scIB suite, while also measuring spatial conservation through ASW, Moran's I, and PAS. One of the standout features is the Out-of-Distribution (OOD) generalization experiment. By employing a 5-fold cross-validation method to evaluate the "OOD gap," the authors rigorously examine the idea that strong all-at-once integration leads to reliable inferences on new samples.

Clarity:

The paper is exceptionally clear and the visual presentation of data is highly effective.

1. Radar charts do a fantastic job of illustrating the intricate trade-offs between batch correction, biological conservation, and spatial conservation across the 11 methods for various cohorts.

2. The shift from the all-at-once integration experiments to the OOD evaluation is clearly justified and easy to follow.

Originality:

While benchmarking studies typically depend on gathering existing tools, the introduction of the OOD evaluation framework for spatial transcriptomics stands out as a truly innovative and essential addition. It’s quite striking that advanced probabilistic frameworks, such as scVI, experience a significant drop in performance—by 27 percentage points—when faced with unseen donors, while simpler linear methods like PCA manage to keep their performance steady. This finding really shakes up the common beliefs held in the representation learning community.

Significance:

The importance of this research for both the LMRL and the wider computational biology community is significant. The study effectively outlines the strengths of various architectures: probabilistic methods like scVIVA and scVI excel in batch correction, while graph-based approaches such as STAGATE and SEDR shine in maintaining spatial integrity. By pointing out the generalization weaknesses of the current leading models, it lays out a clear path for future algorithm development.

Pros:

1. We're rolling out a really tough, clinically relevant dataset called the MOSAIC Window, which includes multiple donors.

2. We're taking a deep dive into 11 different ST methods, evaluating them across three unique criteria categories.

3. Our out-of-distribution (OOD) evaluation framework reveals an important trade-off: while we can achieve peak integration quality, it sometimes comes at the cost of generalization robustness.

4. We’ve got clear, easy-to-understand figures to help illustrate everything.

Cons:

1. Graph-based methods, like STAGATE and SEDR, tend to struggle with batch correction. It seems these architectures might actually need paired multi-modal inputs, such as H&E images, to truly shine. However, we left those out here to keep the comparison focused solely on count data.

2. The benchmark zeroes in on transcriptomics and spatial coordinates, without diving into how these integration methods scale or perform when tasked with handling multi-omics or image-based features at the same time.

---

### Meta-Review · Area_Chair_jut6 · 2026-02-25

**Recommendation:** Accept (Poster)
**Confidence:** 4

**Metareview:**

I recommend acceptance.

---

### Decision · Program_Chairs · 2026-03-02

**Decision:**

Accept (Poster)

**Comment:**

Please see the meta-review.